

# Identification of key genes associated with progression and prognosis for lung squamous cell carcinoma

Xiaohan Ma[1,2,*], Huijun Ren[1,2,*], Ruoyu Peng[1,2], Yi Li[1,2] and Liang Ming[1,2]

[1] Department of Clinical Laboratory, the First Affiliated Hospital of Zhengzhou University, Zhengzhou, Henan, China
[2] Key Clinical Laboratory of Henan Province, Zhengzhou, Henan, China
* These authors contributed equally to this work.

## ABSTRACT

**Background:** Lung squamous cell carcinoma (LUSC) is a major subtype of lung cancer with limited therapeutic options and poor clinical prognosis.
**Methods:** Three datasets (GSE19188, GSE33532 and GSE33479) were obtained from the gene expression omnibus (GEO) database. Differentially expressed genes (DEGs) between LUSC and normal tissues were identified by GEO2R, and functional analysis was employed using the Database for Annotation, Visualization and Integrated Discovery (DAVID) online tool. Protein–protein interaction (PPI) and hub genes were identified via the Search Tool for the Retrieval of Interacting Genes (STRING) and Cytoscape software. Hub genes were further validated in The Cancer Genome Atlas (TCGA) database. Subsequently, survival analysis was performed using the Kapla–Meier curve and Cox progression analysis. Based on univariate and multivariate Cox progression analysis, a gene signature was established to predict overall survival. Receiver operating characteristic curve was used to evaluate the prognostic value of the model.
**Results:** A total of 116 up-regulated genes and 84 down-regulated genes were identified. These DEGs were mainly enriched in the two pathways: cell cycle and p53 signaling way. According to the degree of protein nodes in the PPI network, 10 hub genes were identified. The mRNA expression levels of the 10 hub genes in LUSC were also significantly up-regulated in the TCGA database. Furthermore, a novel seven-gene signature (FLRT3, PPP2R2C, MMP3, MMP12, CAPN8, FILIP1 and SPP1) from the DEGs was constructed and acted as a significant and independent prognostic signature for LUSC.
**Conclusions:** The 10 hub genes might be tightly correlated with LUSC progression. The seven-gene signature might be an independent biomarker with a significant predictive value in LUSC overall survival.

Corresponding author
Liang Ming,
mingliang_2015@sina.com

## INTRODUCTION

Lung cancer, as a highly malignant cancer, is still a common cause for healthy issues worldwide. With approximately 1.8 million deaths in 2018 (*Bray et al., 2018*), lung cancer has been ranked in top two in China and possessed the highest mortality rate (*Sun et al., 2018*). Among all the lung cancer types, non-small cell lung cancer (NSCLC) accounts for over 85% of total cases. Lung squamous cell carcinoma (LUSC) is the second most frequent subtype of NSCLC, accounting for about 40% of NSCLC (*Chen et al., 2014*), and its therapy and prognosis are still facing huge challenges. Nowadays, surgery and adjuvant chemotherapy are the standard treatment for stage I and II of NSCLC, however, molecular analysis is the key to select a first-line therapy for advanced cancer (*Kuo et al., 2019*). Large numbers of gene mutations have been reported to be served as specific biomarkers for diagnosis, treatment and prognosis for LUAD (lung adenocarcinoma) (*Calvayrac et al., 2017*), such as bevacizumab against VEGF. Although complex genomic alterations were found in LUSC, which were different from LUAD, there were no available and specific targeted agents for LUSC until now (*Hirsch et al., 2017*). Despite great progress made in combined therapies, the prognosis of LUSC is still dismal. Hence, there is an imperative and urgent need for identifying key molecules for therapy and prognosis of LUSC.

Recently, high throughput sequencing and microarray technologies have been widely used to investigate relationship between diverse diseases and key molecules, including genes, miRNAs, long non-coding RNAs (lncRNAs) and circRNAs. *Yeo et al. (2017)* proved that programed cell death 1 was over-expressed in LUSC and could be useful for the prediction of poor prognosis. *Hou et al. (2014)* identified 95 up-regulated and 749 down-regulated lncRNAs in response to cisplatin chemo, which indicate that dysregulated lncRNAs are related with therapy and are prognostic markers in LUSC. Several other key molecules were also found involved in the development, diagnosis, and prognosis of LUSC, such as circRNA_103827, circRNA_000122 (*Xu et al., 2018*), peroxiredoxin 4 (*Hwang et al., 2015*), AURKA, BIRC5, LINC00094 (*Li et al., 2017*), and so on. The dysregulation of these mentioned molecules is associated with the progression and prognosis of LUSC; however, limited samples and significant variability among different projects reduce the credibility of these obtained results.

In order to search for promising key genes associated with the progression and prognosis of LUSC, differentially expressed genes (DEGs) in LUSC were identified using three microarray datasets from the Gene Expression Omnibus (GEO) database. Subsequently, ten hub genes were identified by Gene Ontology (GO), Kyoto Encyclopedia of Genes and Genomes (KEGG) enrichment analysis and protein-protein interaction (PPI) network, which were involved in cell cycle and p53 pathway in LUSC. The expression of the ten hub genes was validated in the TCGA database. In addition, a novel seven-gene signature was established to predict effectively overall survival in LUSC.

## MATERIALS AND METHODS

### Data collection

The mRNA expression profiles and corresponding clinical information were acquired from the GEO database (https://www.ncbi.nlm.nih.gov/geo/) and the Cancer Genome Atlas (TCGA) database (https://portal.gdc.cancer.gov). With searching for "lung squamous cell carcinoma", a total of 4,627 series about LUSC were searched from the GEO database. After a careful review, three gene expression profiles (GSE19188, GSE33532 and GSE33479) were collected. The former two databases were both based on GPL570 platform ((HG-U133_Plus_2) Affymetrix Human Genome U133 Plus 2.0 Array), and GSE33479 was based on GPL6480 platform (Agilent-014850 Whole Human Genome Microarray 4 × 44 K G4112F). A total of 57 LUSC tissue samples and 112 normal lung tissue samples were collected from the three GEO datasets. A total of 551 samples were collected from TCGA database, containing 502 LUSC tissues samples and 49 normal tissue samples.

### Data pre-processing and identification of DEGs

In this study, the GEO2R (https://www.ncbi.nlm.nih.gov/geo/geo2r/), an interactive web tool used to compare two groups of samples according to GEO series, was applied to detect the DEGs between 57 LUSC samples and 112 normal samples. The adjusted $P$-value < 0.05 and |log2 (fold change)| ≥ 2.0 were regarded as the cutoff criteria to select the DEGs. Then the DEGs obtained from different GSE datasets were further identified using the Venn diagram web tool (bioinformatics.psb.ugent.be/webtools/Venn/).

### Functional analysis and PPI network construction of DEGs

Gene Ontology and Kyoto Encyclopedia of Genes and Genomes analysis of DEGs were performed using the Database for Annotation, Visualization and Integrated Discovery (DAVID) (https://david.ncifcrf.gov/). The adjusted $P$-value < 0.05 was considered as statistically significant. In order to evaluate protein-protein interactions of DEGs, Cytoscape software and the Search Tool for the Retrieval of Interacting Genes (STRING) (https://string-db.org/) were used to analyze PPI relationship, and PPI pairs with a combined score > 0.4 were extracted. A cytoscape plugin, cytoHubba, was utilized to calculate the degree of protein nodes, and the top rank ten genes were selected as hub genes.

### Hub genes validation and analysis

The mRNA expression profiles in 502 LUSC samples and 49 normal samples from TCGA database were analyzed by the "limma" R package. The statistical analysis was performed using the Wilcox test, and adjusted $P$ < 0.05 and |log2 (fold change)| ≥ 2.0 were selected as the cutoff criteria. Subsequently, the expression levels of hub genes were validated in TCGA database. Besides, the expression levels of the ten hub genes in LUSC were compared with other NSCLC histologic subtypes using the Oncomine database (www.oncomine.org).

The Cox proportional hazards regression model was used for overall survival (OS) analysis among the ten hub genes. Besides, Kaplan–Meier Plotter (http://kmplot.com/) was used to assess the effect of the hub genes on LUSC prognosis. The option "only JetSet best probe set" was selected for probes of genes, and only the LUSC patients were selected for analysis. At last, there were 524 LUSC patients for OS analysis, 141 LUSC patients for FP analysis, and 20 LUSC patients for PPS analysis in the Kaplan–Meier Plotter. $P < 0.05$ was regarded as statistically significant. Then the genes with prognostic values were analyzed to identify their associations with tumor grade using the Oncomine database.

## Survival analysis

In order to identify the potential prognostic values of the DEGs in LUSC, univariate and multivariate Cox regression were conducted using the R package. The clinical information of LUSC patients was downloaded from TCGA database, and 488 LUSC patients were used for analysis after removing the patients (14 of 502) with incomplete clinical data. Only 187 of 200 DEGs could be found and validated in TCGA database, and they were used for the univariate Cox regression analysis. The genes associated with overall survival ($P < 0.05$) were subjected to the multivariate Cox proportional hazards model to establish a robust gene prognostic signature for LUSC. The LUSC patients were further grouped into "high-risk" and "low-risk" based on the median risk score. What's more, a receiver operating characteristic (ROC) curve was constructed to evaluate the predictive accuracy of the gene signature by using the R package "survival ROC". Besides, the gene signature and clinicopathological parameters (age, gender, tumor stage, T/N/M status) were submitted to Cox regression analysis to identify independent factors for OS in LUSC patients.

## RESULTS

### Identification of DEGs in LUSC

All the mRNA expression profiles of GSE19188, GSE33479 and GSE33532 were provided in Table S1–S3. According to the criteria of adjusted $P$-value $< 0.05$ and |log2 (fold change)| $\geq 2$, a total of 458 up-regulated genes and 668 down-regulated genes were identified in GSE19188 (Fig. 1A). From GSE33479, 1153 DEGs were identified, including 499 up-regulated genes and 654 down-regulated genes (Fig. 1B). From GSE33532, 1375 DEGs including 569 up-regulated genes and 806 down-regulated genes were identified (Fig. 1C). All the DEGs were identified by comparing LUSC samples and normal lung samples. Subsequently, the results from the three studies were analyzed using the Venn diagram tool, and 116 up-regulated genes and 84 down-regulated genes were identified (Figs. 1D and 1E; Table S4).

### Functional analysis and PPI network construction

GO and KEGG pathway enrichment analysis of DEGs were carried out to explore the biological functions of 200 DEGs (Table 1). The enriched GO terms of DEGs were classified into three categories: molecular functions (MF), cellular components (CC), and biological processes (BP). As shown in Fig. 2, the DEGs were mainly enriched in
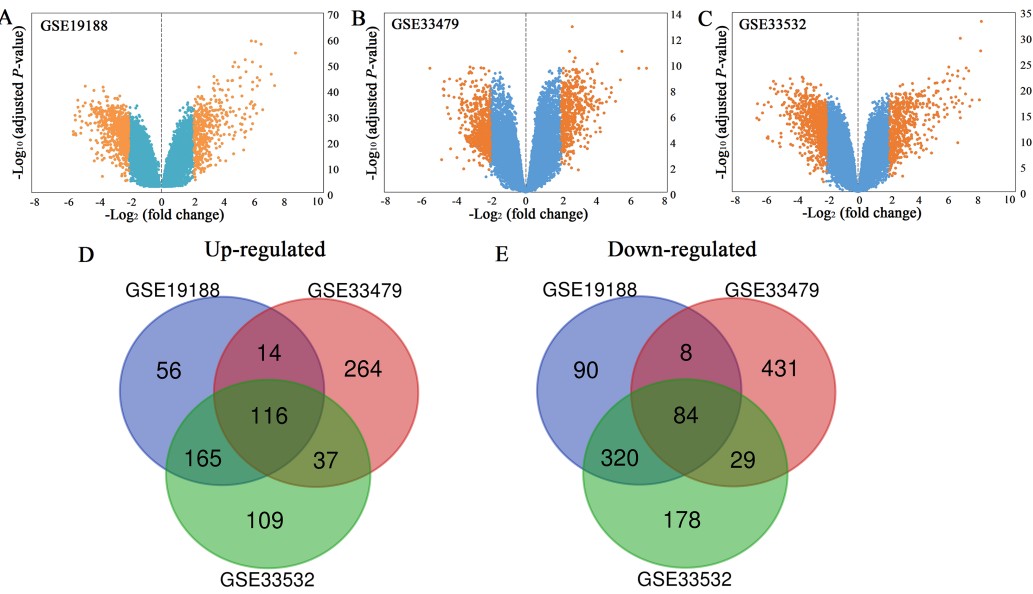

**Figure 1 Volcano plots (A–C) and Venn diagrams (D and E).** (A–C) The DEGs of GSE19188, GSE33479 and GSE33532. The orange dots represent DEGs filtered based on the cutoff criteria of adjusted *P*-value < 0.05 and |log2 (fold change)| ≥ 2. While the blue dots represent genes that are not satisfied the cutoff criteria. The genes with log2 (fold change) ≥ 2 were up-regulated and the genes with log2 (fold change) ≤ −2 were down-regulated. (D and E) The common up-regulated and down-regulated DEGs in the three datasets.                                   

"mitotic nuclear division", "cell division", and "sister chromatid cohesion" in the BP category. Within the CC category, "condensed chromosome kinetochore", "chromosome", and "midbody" were predominant. The results of KEGG analysis exhibited that the DEGs were mainly involved in "Cell cycle" and "p53 signaling pathway".

PPI analysis could reflect the molecular mechanisms of cancer physiological and pathological changes. The PPI network of common DEGs was constructed using the STRING V11.0 tool and Cytoscape software (Fig. 3), which containing 199 nodes and 2,363 edges. Subsequently, the top ten genes with high node degree were selected as the hub genes for further analysis (Fig. 4). Results showed that the top ten hub genes were significantly up-regulated in LUSC tissues. Among them, cyclin-dependent kinases 1 (CDK1) was the most significant gene with connectivity degree = 57, followed by BUB1 mitotic checkpoint serine/threonine kinase (BUB1; degree = 48), cyclin B1 (CCNB2; degree = 48), cyclin A2 (CCNA2; degree = 46), kinesin family member 2C (KIF2C; degree = 41), aurora kinase B (AURKB; degree = 41), kinesin family member 11 (KIF11; degree = 41), mitotic arrest deficient 2 like 1 (MAD2L1; degree = 40), DNA topoisomerase II alpha (TOP2A; degree = 40), and DLG associated protein 5 (DLGAP5; degree = 39).

## Hub genes validation and analysis

The expression of the ten hub genes was validated in TCGA database, which was consistent with the results from the GEO database (Fig. 5). The Analysis indicated that their expression levels in LUSC were significantly higher than those in other NSCLC (Fig. 6).

**Table 1 Significantly enriched GO terms and KEGG pathways of DEGs.**

| Category | Term | Description | Count | FDR |
|---|---|---|---|---|
| BP | GO:0007067 | Mitotic nuclear division | 27 | 8.64E−15 |
| BP | GO:0051301 | Cell division | 28 | 4.98E−12 |
| BP | GO:0007062 | Sister chromatid cohesion | 15 | 1.60E−08 |
| BP | GO:0007059 | Chromosome segregation | 12 | 4.09E−07 |
| BP | GO:0000082 | G1/S transition of mitotic cell cycle | 12 | 3.40E−05 |
| BP | GO:0006260 | DNA replication | 11 | 0.01728675 |
| BP | GO:0008283 | Cell proliferation | 16 | 0.029764593 |
| BP | GO:0000086 | G2/M transition of mitotic cell cycle | 10 | 0.041628856 |
| CC | GO:0000777 | Condensed chromosome kinetochore | 14 | 9.87E−09 |
| CC | GO:0000775 | Chromosome, centromeric region | 11 | 5.29E−07 |
| CC | GO:0030496 | Midbody | 14 | 1.54E−06 |
| CC | GO:0000776 | Kinetochore | 10 | 2.51E−04 |
| CC | GO:0005819 | Spindle | 11 | 8.59E−04 |
| CC | GO:0005578 | Proteinaceous extracellular matrix | 15 | 0.00144238 |
| CC | GO:0005654 | Nucleoplasm | 55 | 0.006300472 |
| CC | GO:0005829 | Cytosol | 61 | 0.01409558 |
| MF | GO:0003777 | Microtubule motor activity | 10 | 2.36E−04 |
| KEGG | hsa04110 | Cell cycle | 12 | 1.59E−05 |
| KEGG | hsa04115 | p53 Signaling pathway | 7 | 0.037570687 |

**Note:**
BP, biological process; CC, cellular component; MF, molecular function; DEG, differentially expressed gene; GO, Gene Ontology; KEGG, Kyoto Encyclopedia of Genes and Genomes; FDR, adjust *P* value.

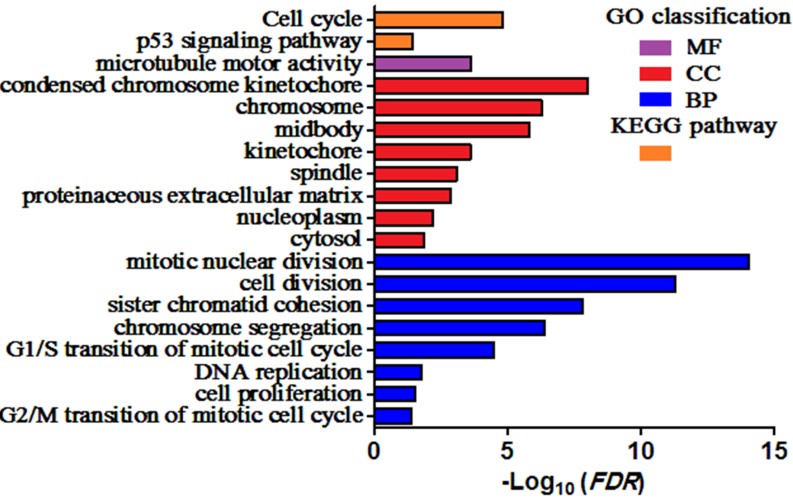

**Figure 2 Enriched GO terms and KEGG pathways of the DEGs.** MF, molecular function; CC, cellular component; BP, biological process. Adjusted *P* value (FDR) < 0.05 was considered significant.

Cox regression model was used to analyze the ten hub genes, however, none of them were associated with OS, and no significant gene signature could be established to predict OS in LUSC patients. Kaplan–Meier Plotter showed that the up-regulated CCNA2, KIF11,

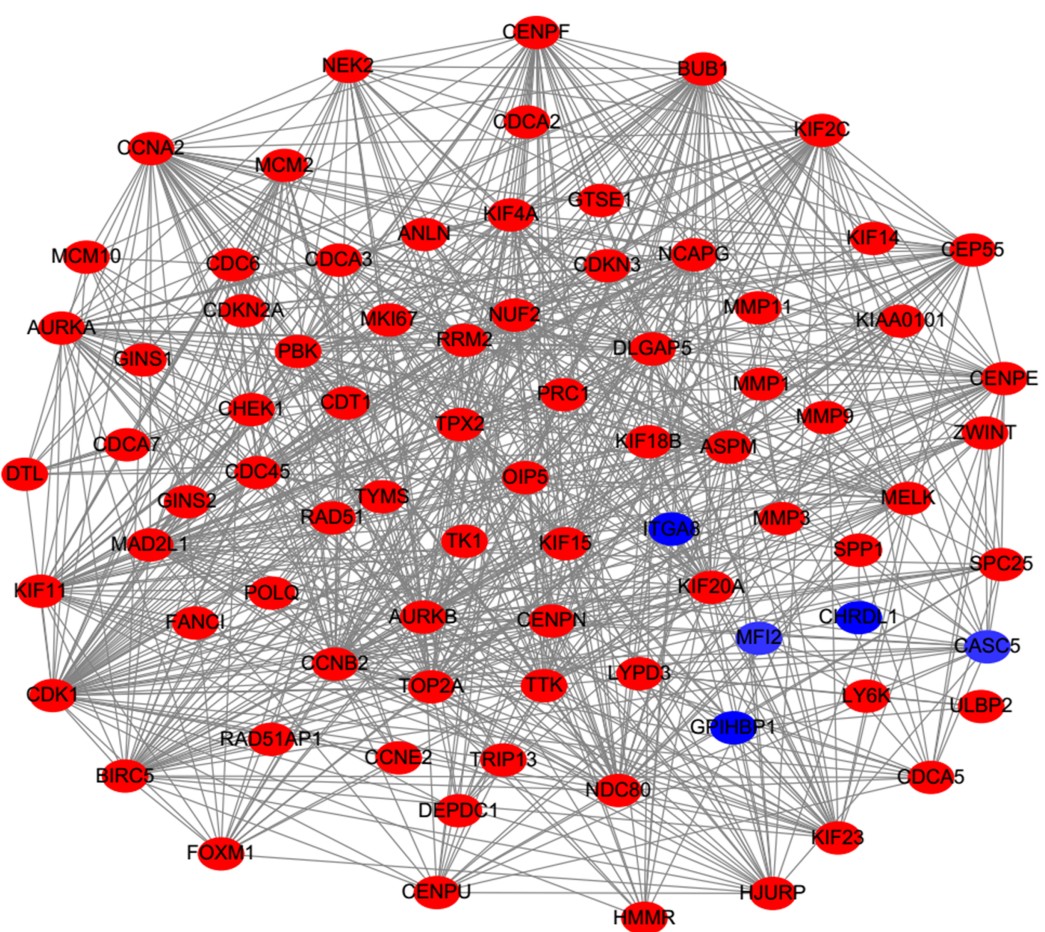

**Figure 3 Protein–protein interaction network of DEGs.** The red nodes represent the up-regulated DEGs. The blue nodes represent the down-regulated DEGs.

MAD2L1, and DLGAP5 were all related to worse first-progression survival (FP) in LUSC patients (Figs. 7A–7D). Only the up-regulation of KIF2C was associated with unfavorable post-progression survival (PPS) in LUSC patients (Fig. 7E). Among the five hub genes with prognostic values genes, four (CCNA2, DLGAP5, MAD2L1 and KIF2C) were associated with LUSC tumor grade (Fig. 8).

## Cox progression analysis and construction of prognostic signature

Only 187 of 200 DEGs were found and validated in TCGA database. Then the 187 DEGs were used for the univariate Cox regression analysis, and 23 of 187 DEGs were significantly associated with the OS ($P < 0.05$), as shown in Table S5. Subsequently, a seven-gene prognostic signature was established by the multivariate Cox regression analysis (Table 2). Risk Score = 0.0341 × expression of FLRT3 + 0.0339 × expression of PPP2R2C + 0.0058 × expression of MMP3 + 0.0031 × expression of MMP12 + 0.0518 × expression of CAPN8 + 0.1323 × expression of FILIP1 + 0.0002 × expression of SPP1. According to the signature, risk score was calculated for each patient and each patient was grouped into
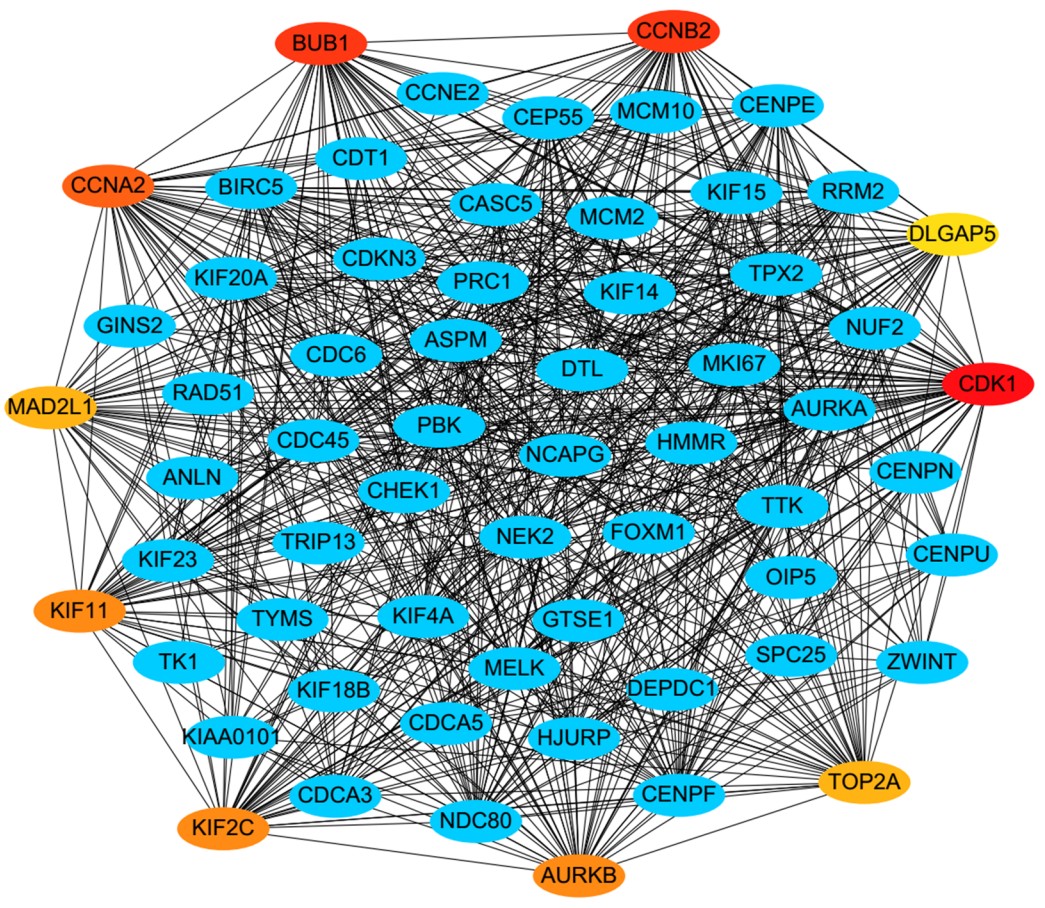

**Figure 4  A PPI network for the hub genes identified with the Cytoscape.** The top ten hub genes were ranked by the degree, and advanced ranking is represented by a redder color.

"high-risk" and "low-risk" according to the median risk score. The distribution of signature risk score (Fig. 9A), survival status and survival time (Fig. 9B) in the model group were presented. The expression of the seven genes in high and low-risk groups was shown in Fig. 9C. Compared with the low-risk patients, the high-risk patients presented worse OS (Fig. 9D). According to the ROC curve on 5-year OS (Fig. 9E), an area under curve (AUC) value of 0.707 (>0.7) indicated that the prognostic prediction value of the seven-gene signature was reliable.

## Independence assessment of seven-gene mRNA signature

Univariate Cox progression analysis indicated that risk score, age, tumor stage and T status were significantly associated with the OS among LUSC patients. Subsequently, multivariate Cox analysis showed that the gene signature had a significantly independent prognostic value with $P < 0.001$ (Table 3). These results revealed that the seven-gene risk signature was an independent prognostic factor for LUSC after adjusting the confounding effects.

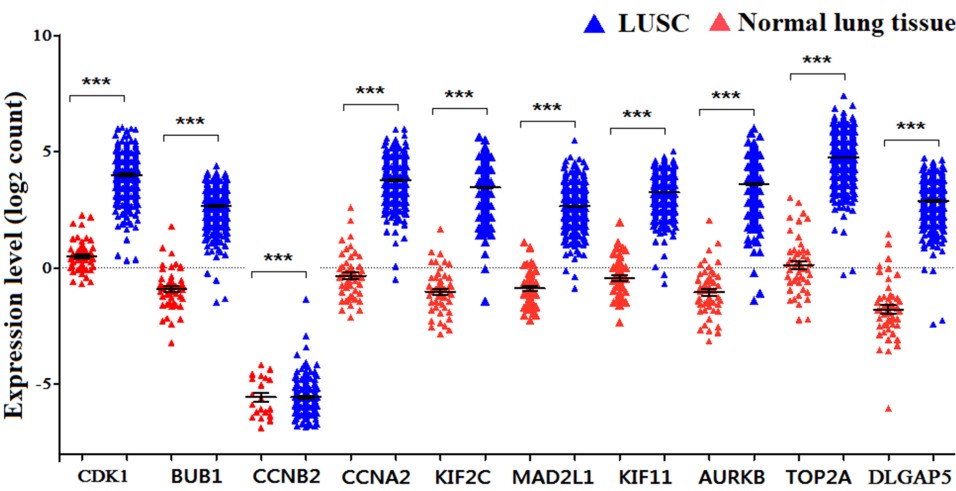

**Figure 5 The expression levels of 10 hub genes in the TCGA database.** The expression of the ten hub genes in 502 LUSC tissues and 49 normal tissues from TCGA database. Gene expression values are log₂-transformed.

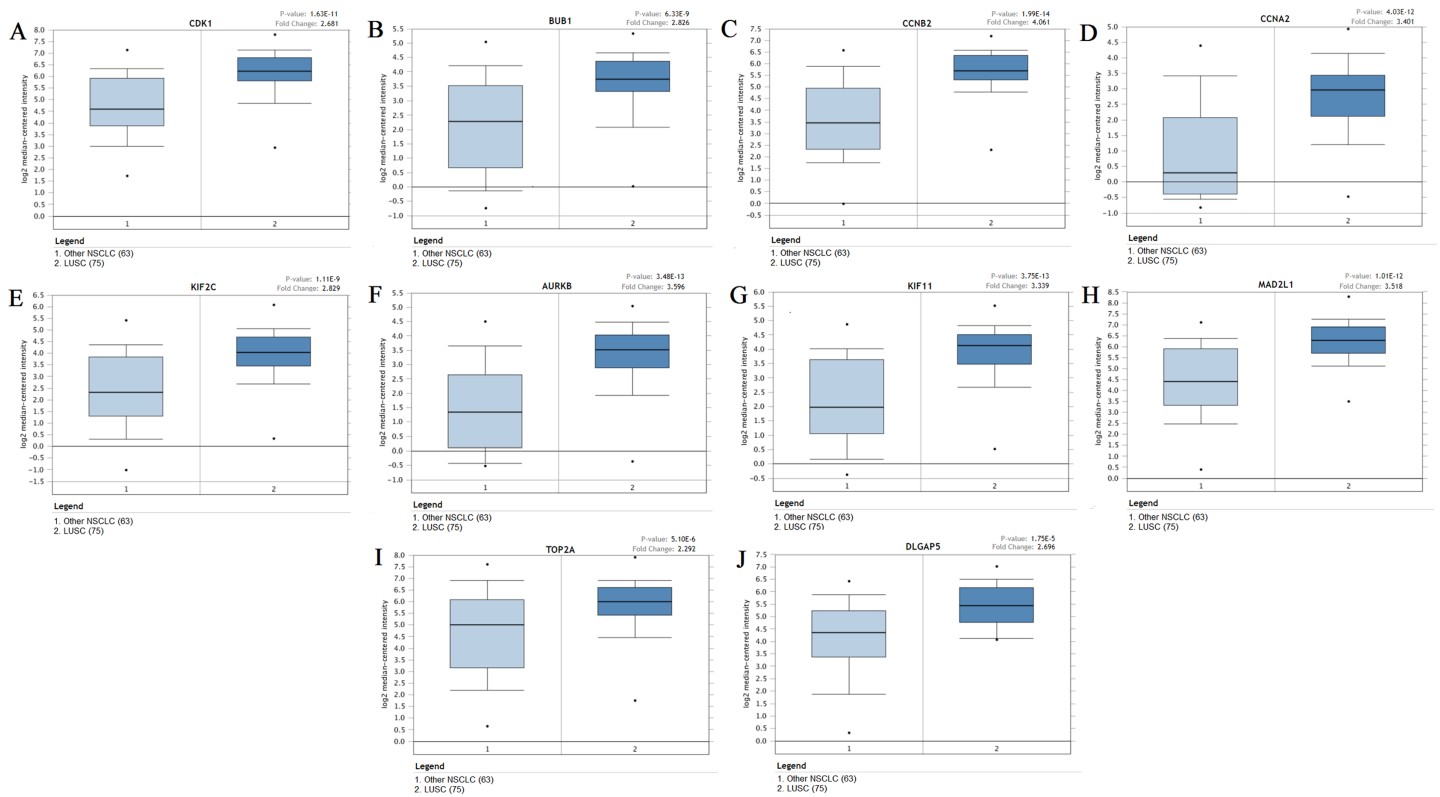

**Figure 6 The expression levels of 10 hub genes between different subtypes.** (A–J) The expression of the ten hub genes (CDK1, BUB1, CCNB2, CCNA2, KIF2C, AURKB, KIF11, MAD2L1, TOP2A, DLGAP5) in LUSC and other subtypes. Their expression in LUSC was significantly different from that in other subtypes, which was analyzed using Oncomine database. Gene expression values are log2-transformed.

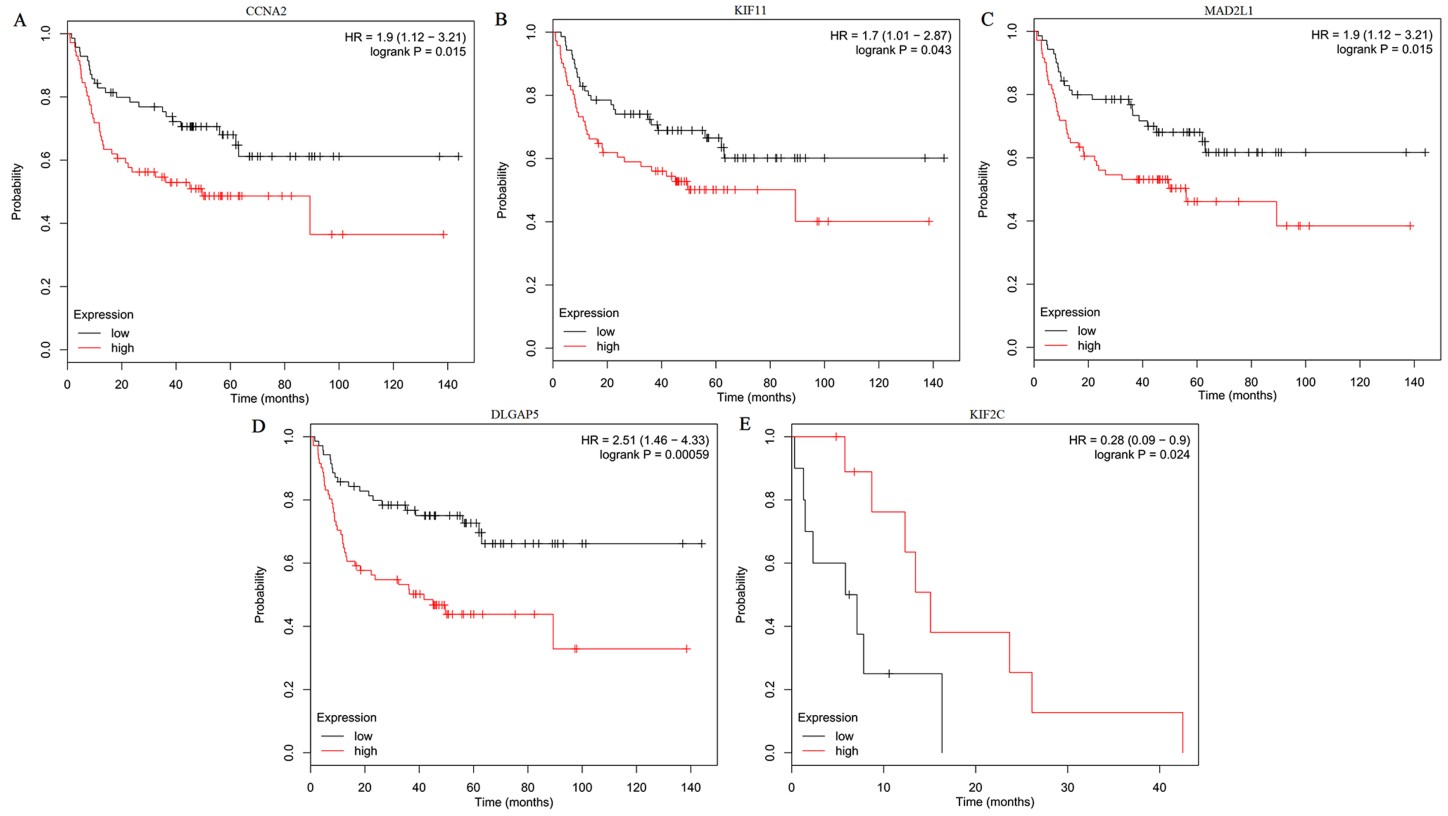

**Figure 7 Kaplan–Meier survival analyses of hub genes in LUSC patients.** (A–D) The first progression survival analyses of the hub genes (CCNA2, KIF11, MAD2L1 and DLGAP5) were performed. (E) The post progression survival analyses of the hub gene (KIF2C) were performed. Logrank $P < 0.05$ was considered statistically significant.

## DISCUSSION

In this study, a total of 116 up-regulated DEGs and 84 down-regulated DEGs were identified from the GEO database. KEGG enrichment analysis showed that these DEGs were mainly involved in two pathways: "Cell cycle" and "p53 signaling pathway". Increasing evidences suggest that disordered cell cycle has been a mark of tumors (*Hanahan & Weinberg, 2011*). In LUSC, cell cycle progression and cell proliferation could be inhibited by CCNB1 (*Wang et al., 2019*), promoted by DDA1 (*Cheng et al., 2017*), and arrested in the G phase by SART3 (*Sherman, Mitchell & Garner Amanda, 2019*). Besides, the cell cycle is closely related with the p53 pathway (*Joerger & Fersht, 2016*). P53, a multifunctional transcription factor in cancer progression, participates in the regulation of cell cycle, metabolic pathways, and so on (*Stegh, 2012*; *Fridman & Lowe, 2003*; *Vogelstein et al., 2000*). Studies have found that p53 signaling pathway could induce cancer cell apoptosis by targeting at Bax (*Jin et al., 2017*), p21, and HIF1 α (*Yang et al., 2018*) in LUSC. Besides, p53 plays the role in tumor suppression in LUSC via being regulated by other genes, such as miR-223-3p (*Luo et al., 2019*), which has not been found in the studies of other cancers. In short, the enriched pathways of these DEGs could interact with each other and participate in the regulation of LUSC progression.

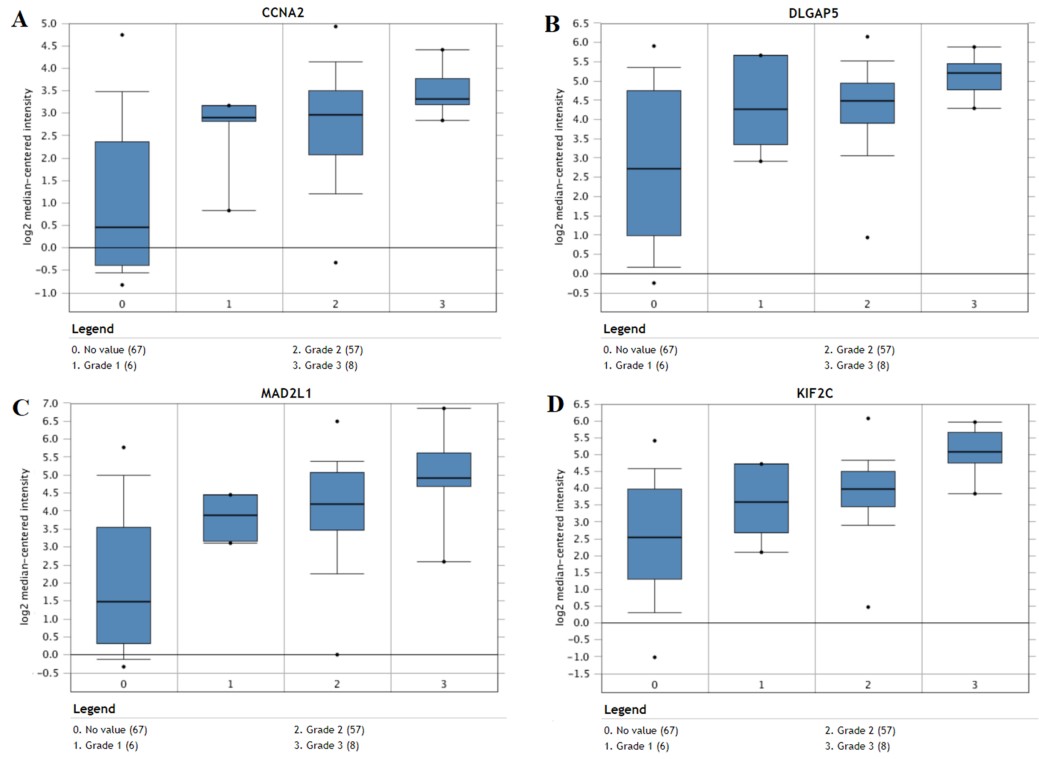

**Figure 8 The association between the expression of prognosis-related hub genes and tumor grade (A–D).** In the Lee Lung dataset, the altered CCNA2, DLGAP5, MAD2L1 and KIF2C were associated with tumor grade in the LUSC progression. 0: No value; 1: Grade 1: high differentiation; 2: Grade 2: middle differentiation; Grade 3: poor differentiation.

**Table 2 Prognostic values for the seven-genes prognostic signature in 488 LUSC patients.**

| Id | Coef | HR | HR.95L | HR.95H | *P* value |
|---|---|---|---|---|---|
| FLRT3 | 0.034099635 | 1.034687693 | 1.004943783 | 1.065311951 | 0.021943406 |
| PPP2R2C | 0.033850795 | 1.034430253 | 1.000444568 | 1.069570452 | 0.047028752 |
| MMP3 | 0.005741905 | 1.005758421 | 1.000626781 | 1.010916378 | 0.027803799 |
| MMP12 | 0.003057819 | 1.003062498 | 1.000893676 | 1.00523602 | 0.005626057 |
| CAPN8 | 0.051756734 | 1.053119523 | 1.013625297 | 1.094152576 | 0.007956628 |
| FILIP1 | 0.132257889 | 1.141402636 | 1.038057415 | 1.255036532 | 0.006308144 |
| SPP1 | 0.0002455 | 1.00024553 | 1.000013897 | 1.000477217 | 0.037749151 |

In order to predict the function associations between the 200 DEGs, a PPI network was constructed and ten hub genes were identified, including CDK1, BUB1, CCNB2, CCNA2, KIF2C, AURKB, KIF11, MAD2L1, TOP2A, and DLGAP5. Survival analysis revealed that five of hub genes were associated with FP or PPS. Among the five hub genes, the altered CCNA2, DLGAP5, MAD2L1 and KIF2C were associated with tumor grade, and were significantly up-regulated in LUSC compared with other NSCLC, implicating crucial roles of them in LUSC progression.

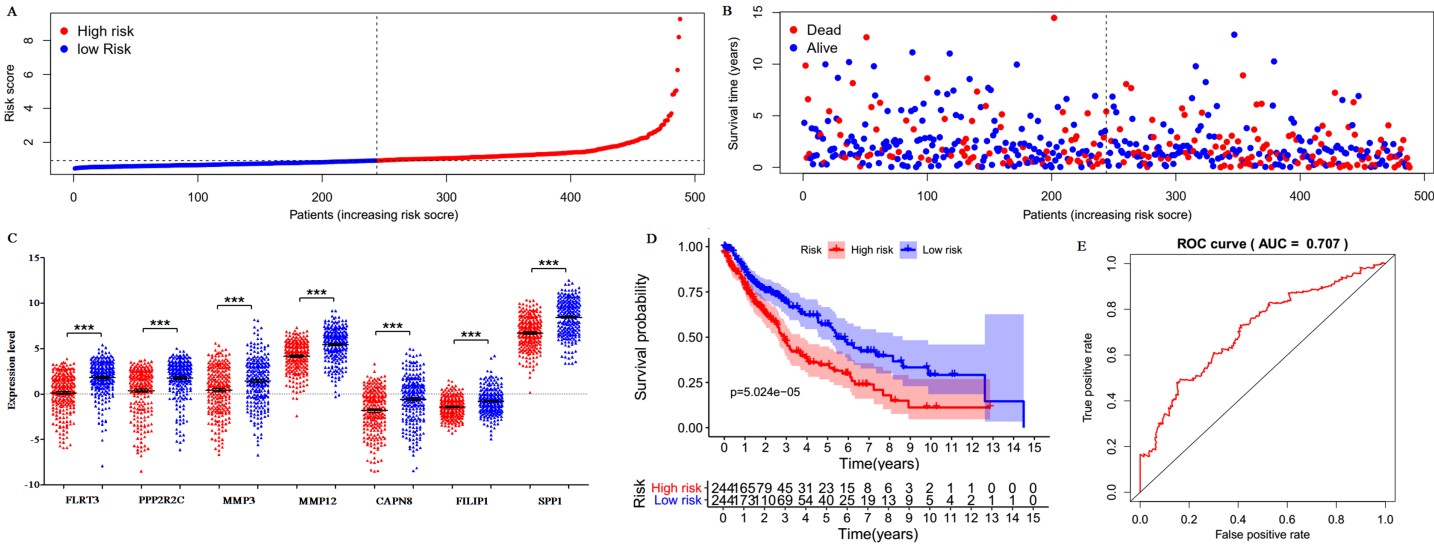

**Figure 9 Prognostic gene signature of the seven genes in 488 LUSC patients.** (A) Risk score distribution. (B) Patients' survival status distribution. (C) Expression levels of the seven genes in low and high-risk groups (TCGA database). Gene expression values are log2-transformed. (D) The survival curves of LUSC patients in high- and low-risk groups. (E) Time-dependent ROC curves for predicting OS in LUSC patients by the risk score.

**Table 3 Univariate and multivariate Cox regression analysis of OS in LUSC.**

| Clinical feature | Univariate analysis | | | Multivariate analysis | | |
|---|---|---|---|---|---|---|
| | HR | 95% CI | *P*-value | HR | 95% CI | *P*-value |
| Risk score (high/low) | 1.655 | [1.431–1.913] | <0.001 | 1.642 | [1.412–1.412] | <0.001 |
| Age (year) | 1.024 | [1.004–1.045] | 0.02 | 1.028 | [1.006–1.049] | 0.011 |
| Gender (male/female) | 1.263 | [0.863–1.849] | 0.229 | 1.289 | [0.878–1.890] | 0.195 |
| Stage (IV, III/II, I) | 1.261 | [1.044–1.524] | 0.016 | 0.982 | [0.636–1.518] | 0.936 |
| T (T4, T3/T2, T1) | 1.311 | [1.059–1.623] | 0.013 | 1.272 | [0.925–1.749] | 0.138 |
| M (M1/M0) | 1.829 | [0.581–5.761] | 0.302 | 1.816 | [0.455–7.253] | 0.399 |
| N (N3, N2, N1/N0) | 1.159 | [0.93–1.442] | 0.189 | 1.182 | [0.802–1.742] | 0.398 |

CCNA2, also known as CyclinA2, a highly conserved cyclin protein (*Ko et al., 2013*), has been found significantly over-expressed in various cancers (*Gao et al., 2014*). Previous researches showed that CCNA2 could regulate cell cycle in cancers via controlling the G1/S and G2/M transitions (*Arsic et al., 2012*). CCNA2 is also associated with epithelial-mesenchymal transition and cancer progression (*Bendris et al., 2012*). Previous studies have shown that diverse molecules, such as coiled-coil domain containing 6, Zinc finger SWIM-type containing 5, and miR-137, could regulate the proliferation, invasion, and migration of NSCLC cells by changing the expression of CCNA2 (*Morra et al., 2015*; *Xu et al., 2018*; *Chen et al., 2017*). In the present study, CCNA2 was found up-regulated in LUSC tissues compared with normal tissues, and over-expression of CCNA2 was related to worse first-progression survival in LUSC patients. These results

indicated that CCNA2 might be a progression biomarker and prognostic indicator for LUSC.

Kinesin family member 11 (KIF11), a kinesin-5-family protein, could affect tumor development by controlling the correct arrangement of the microtubules, which was the key stage in mitosis (*Blangy et al., 1995*). Previous studies have reported that KIF11 was up-regulated in lung cancer tissues compared with normal tissues, and associated with poor overall survival (*Al-Khafaji et al., 2017*; *Schneider et al., 2017*). *Kato et al. (2018)* declared that high-level KIF11 was as an independent prognostic factor in LUSC, and might be promising therapeutic option for advanced lung cancer. In this study, KIF11 was both identified as one of over-expressed hub genes and related to unfavorable first-progression survival of LUSC patients, which was consistent with previous studies.

MAD2 mitotic arrest deficient-like 1 (MAD2L1), an important spindle checkpoint protein, plays important roles in protecting cells from abnormal chromosome segregation (*Yu, 2006*). MAD2L1 has been shown high expression levels and might be a biomaker for poor prognosis in various cancers, such as breast cancer (*Wang et al., 2015*), osteosarcoma (*Sun, Li & Yan, 2015*), and so on. MAD2L1 has been identified as a potential therapeutic target gene in NSCLC (*Zhou et al., 2015*), and could be a promising prognostic biomarker for LUAD (*MacDermed et al., 2010*) and small cell lung cancer (*Liao et al., 2019*), however, there was no similar results in LUSC. In this study, over-expression of MAD2L1 and its prognostic value in LUSC were identified.

Kinesin family member 2C (KIF2C) has been served as a modulator in bipolar spindle formation, microtubule depolymerization, and chromosome segregation, and it could promote the tumor proliferation and metastasis (*Gan et al., 2019*). In this study, down-regulated KIF2C had a significant value in post-progression survival in LUSC. These suggest that KIF2C might be involved in tumor progression and be a potential prognostic factor in LUSC. Disc large (drosophila) homolog-associated protein 5 (DLGAP5), as an important mitotic spindle protein, participated in cancers development and progression (*Liao et al., 2013*). DLGAP5 has been found significantly over-expressed in different subtypes of lung cancer, and it could be a promising prognostic biomarker and therapeutic target (*Qi et al., 2019*), which are consistent with the results of this study.

Proteins encoded by CDK1, BUB1 and AURKB belong to the serine/threonine kinases family, and overexpression of them has been detected involved in various tumors prognosis via regulating tumor cell cycle. CDK1 (*Xie et al., 2019*) and BUB1 (*Piao et al., 2019*) have been proved to influence tumor progression by inducing cell cycle dysregulation. AURKB was proved to be a therapeutic target by triggering G1/S arrest in NSCLC (*Bertran-Alamillo et al., 2019*). CCNB2, encoding the member of the cyclin family, could promote tumor progression by facilitating cell proliferation and maintain normal G2/M transition. CCNB2 and BUB1 might be involved in the cancer stem cells to promote LUSC progression (*Qin et al., 2020*). TOP2A encoding a DNA topoisomerase, could affect overall survival and clinicopathological features in NSCLC by altering the transcription of DNA (*Hou et al., 2017*). Much further investigations on the roles of the ten genes in LUSC are needed. In this report, functional analysis and their roles in other tumors might provide valuable clues to investigate their roles in LUSC progression.

In this study, none of the ten hub gene was found associated with OS in LUSC by K–M plotter analysis and cox progression analysis using TCGA database. Hence, cox progression analysis was performed for all the DEGs. Compared to a single gene marker to predict patient survival, a gene signature will provide a stable and effective prediction effect. Therefore, a novel seven-gene signature (including FLRT3, PPP2R2C, MMP3, MMP12, CAPN8, FILIP1, and SPP1) was established for LUSC prognostic prediction. According to the gene signature, each LUSC patient with a risk score was classified as high-risk or low-risk. The risk scores of high-risk patients were much higher than that of low-risk patients, which proved that the risk score was significant. Results showed that high-risk patients presented significantly worse OS than low-risk patients. ROC demonstrated that the seven-gene signature was efficient and sensitive in the survival prediction in LUSC. In addition, clinical pathological features (including age, gender, grade, and T/M/N state) are known prognostic factors in LUSC. In order to avoid the interference effect of these known prognostic factors, multivariate Cox regression analysis was carried out, and proved that the signature was an independent prognostic factor in LUSC. All these results suggested that the signature could be an efficient and independent indicator for LUSC prognosis.

*Bluemn et al. (2013)* demonstrated the loss of PPP2R2C was associated with cancer recurrence and a poor survival in prostate cancer. Nevertheless, the role of PPP2R2C in LUSC remains unclear. Previous studies have proved MMP3 (matrix metalloproteinase 3) played crucial roles in invasion and metastasis in many cancers (*Ma et al., 2019*). MMP3 is associated with poor survival in various cancers, and its polymorphism might increase the risk of lung cancer (*Hu et al., 2013*). Up-regulated MMP12 has been found associated with the pathological stage and tumor metastasis in lung cancer (*Lv et al., 2015*). CAPN8 has been proved up-regulated in lung cancer and could be a prognostic biomarker (*Zhang et al., 2015*). FILIP1L (FILamin A Interacting Protein 1-Like) is involved in cell proliferation and migration by inhibiting the WNT signaling pathway (*Kwon et al., 2014*). Up-regulated SPP1 (secreted phosphoprotein 1) could promote cell proliferation, migration, and invasion by Integrin β1/FAK/AKT pathway (*Zeng et al., 2018*), and could be a prognostic indicator in many tumors (*Choe et al., 2018*). Inhibition of SPP1 could enhance the invasion and might be a promising target for NSCLC therapy (*Wang et al., 2019*). To our knowledge, the seven-gene signature for prognostic prediction in LUSC has not been reported previously, and has been demonstrated as an independent and useful prognostic signature in LUSC.

In addition, in the study, many more genes are specific to one database but not the others. This might be caused by the heterogeneity between different datasets. Various factors might lead to the heterogeneity, such as experimental methods and conditions, detection platform and system, sample sources, reagent batches, and so on. Some aspects in the present study have been considered to reduce the influences. Firstly, all of the samples in this study are from normal lung tissues and LUSC tissues, and not from adjacent tissues, which ensures the consistency of the experimental grouping. Secondly, all the data from the GEO database have been normalized, which could reduce variation between groups and make them comparable. Lastly, the DEGs were identified from each

dataset, and then the common part of them were selected as the final DEGs, which could reduce the influences resulted from the heterogeneity of the different datasets. These aspects could make the analysis results more effective and objective.

## CONCLUSIONS

In summary, this study identified 200 DEGs between LUSC and normal tissues by three GEO datasets. Ten up-regulated hub genes were validated in TCGA database and were associated with cell cycle and p53 signaling pathway. CCNA2, DLGAP5, MAD2L1, and KIF2C were significantly up-regulated compared to other subtypes, and associated with tumor stage in LUSC, suggesting that they might tightly be involved in progression and prognosis of LUSC. In addition, a novel seven-gene signature was established to predict overall survival in LUSC, which may help to provide clues in LUSC prognosis.

## ACKNOWLEDGEMENTS

The authors would like to thank Dr. Shan Shan Li, who is affiliated with the CapitalBio Technology Company (Beijing), for her technical assistance.

### Funding

This work was supported by the National Natural Science Foundation of China (No. 81401682). The funders had no role in study design, data collection and analysis, decision to publish, or preparation of the manuscript.

### Grant Disclosures

The following grant information was disclosed by the authors:
National Natural Science Foundation of China: 81401682.

### Competing Interests

The authors declare that they have no competing interests.

### Author Contributions

- Xiaohan Ma conceived and designed the experiments, performed the experiments, analyzed the data, prepared figures and/or tables, authored or reviewed drafts of the paper, and approved the final draft.
- Huijun Ren performed the experiments, analyzed the data, prepared figures and/or tables, authored or reviewed drafts of the paper, and approved the final draft.
- Ruoyu Peng performed the experiments, authored or reviewed drafts of the paper, and approved the final draft.
- Yi Li analyzed the data, prepared figures and/or tables, and approved the final draft.
- Liang Ming conceived and designed the experiments, authored or reviewed drafts of the paper, and approved the final draft.

## Data Availability

The raw measurements are available in Tables S1–S5. The expression of all the genes used for this study are available at GEO datasets: GSE19188, GSE33479, GSE33532. All the results from Cox progression analysis are available in Table S5.

## Supplemental Information

Supplemental information for this article can be found online at http://dx.doi.org/10.7717/peerj.9086#supplemental-information.

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
