# Peer review of "Identification of key genes associated with progression and prognosis for lung squamous cell carcinoma"

_PeerJ, doi:10.7717/peerj.9086_

## Round 0.1 · original submission · Major Revisions

As you can see from the attached reports, the reviewers suggested adding a few additional analyses to the manuscripts as well as adding further clarifications to your manuscript to improve it.

Reviewer 1 ·

Basic reporting

The basic quality is OK.

Experimental design

The design is generally OK. Some methods not so reasonable (see below).

Validity of the findings

The PPI network is on protein level, whereas differential expression is based on mRNA level. It is not reasonable to put differential genes onto the PPI network to identify hub proteins.

Additional comments

1) People rarely say "first and post progression survival", please explain.
2) Need to validate the discovery with the TCGA LUSC dataset.

Reviewer 2 ·

Basic reporting

In the manuscript” Identification of key candidate prognostic genes associated with lung squamous cell carcinoma by bioinformatics analysis” authors have used three datasets and identified the differentially expressed genes. Authors have also identified the PPI network and the prognostic capability of the selected genes.

The manuscript is written in good english.
More references can be provided.

Experimental design

The experiments design for differential expression analysis is okay. However, for the prognostic analysis, a better method can be applied.

Validity of the findings

Authors should have used more data for example TCGA to perform better analysis.

Additional comments

Following are my comments-
1. Figure 1 and Figure 2 are describing the same experiment. Please merge the figure.
2. In figure 2, many more genes are specific to the data set compared to common genes. Authors should comment on this. Is there a possibility that origin of samples, method of analysis etc play a more important role than the disease itself?
3. Authors have suggested that p53 signalling pathway is enriched in the GO analysis. However, it’s not shown in figure 3. Also, these enriched pathways are generally present in the majority of cancer type. Author should discuss in detail how these pathways are different regulated in LUSC compared to other tumor types.
4. Are these significant 10 genes specific to LUSC? I think that the role of these genes is already reported in many cancer types.
5. Authors should also explain how these 10 genes significant for LUSC pathology?
6. figure 6 shows that 10 selected genes are overexpressed in LUSC compared to normal. I think it’s an expected result from this study. Also, these genes are reported to play a critical role in carcinogenesis of many tumor types.
7. Based on my experience, the cox regression is a better method of finding the prognostic markers. Also, why did authors have used only 10 genes for prognostic identification? The genes without differential expression in normal and tumour can be a good prognostic marker. I would suggest authors to perform the analysis with all the genes.
8. For prognostic analysis, which sample set was used and how many patients were used for the analysis.
9. TCGA has around 500 patient’s samples which may provide better understating with above analysis.

·

Basic reporting

Basic reporting of the manuscript is overall flawless. But I recommend the authors to check for typos one final time.
1. Typo in line 90: Date pre-processing --> Data pre-processing

Experimental design

1. Cox regression for potential prognostic factors
I recommend the authors perform a Cox multivariate survival analysis for the potential prognostic factors they have found and discuss about the results. Although the Kaplan-Meier plot results truely suggest that overexpression of either of the 5 candidate genes associate with poor prognosis, survival analysis with known prognostic factors should be given to state the prognostic and/or treatment potential of the candidate genes. But I do not insist that the authors should exclude the candidates that are either correlated with themselves or turn out not to be significant by the Cox regression; presenting the data will be enough for the readers to figure out whether the candidate genes have additional prognostic values.

2. Stating the difference or similarity of the array platforms.
Since the authors used data from three studies spanning two array platforms. I suggest that the authors show the expression histogram of the three respective studies used for data collection as a supplementary figure. Also, the authors should state how different the data they used are, and/or why it does not matter to use the pooled data.

Validity of the findings

Findings of the manuscript seem valid and robust.

---

## Round 0.2 · accepted · Accept

All of the reviewers' comments were addressed with additional analyses and references in the rebuttal letter and revised manuscript.

Reviewer 1 ·

Basic reporting

OK

Experimental design

Valid

Validity of the findings

Valid

Additional comments

All my previous questions have been addressed.